# Measuring Factor XIII Inhibitors in Patients with Factor XIII Deficiency: A Case Report and Systematic Review of Current Practices in Japan

**DOI:** 10.3390/jcm11061699

**Published:** 2022-03-18

**Authors:** Shiho Amano, Kohei Oka, Yutaka Sato, Chiaki Sano, Ryuichi Ohta

**Affiliations:** 1Community Care, Unnan City Hospital, Unnan 699-1221, Japan; amano_shiho95@yahoo.co.jp; 2Department of Community Medicine Management, Faculty of Medicine, Shimane University, Izumo 693-8501, Japan; sanochi@med.shimane-u.ac.jp; 3Medical Department, Shimane University, 89-1 Enya cho, Izumo 693-8501, Japan; kohei313131@outlook.jp (K.O.); m161043@med.shimane-u.ac.jp (Y.S.)

**Keywords:** factor XIII deficiency, bleeding disorder, factor XIII inhibitors, acquired factor XIII deficiency

## Abstract

Factor XIII (FXIII) deficiency is a rare but serious coagulopathy. FXIII is critical in blood coagulation, and FXIII deficiencies can lead to uncontrolled or spontaneous bleeding. FXIII deficiencies can be congenital or acquired; acquired FXIII deficiency can be categorized as autoimmune and non-autoimmune. Immunological tests to measure FXIII inhibitors are required to diagnose acquired FXIII deficiency; however, appropriate test facilities are limited, which increases the turnaround time of these tests. In the case of critical bleeding, delayed test results may worsen prognosis due to delayed treatment. Here, we report a case of acquired FXIII deficiency, followed by a review of FXIII deficiency cases in Japan. We performed a systematic review to investigate the present conditions of the diagnosis and treatment of FXIII deficiency, including the measurement of FXIII inhibitors in Japan. FXIII inhibitor testing was only performed in 29.7 of acquired FXIII deficiency cases. Clinical departments other than internal medicine and pediatrics were often involved in medical treatment at the time of onset. Therefore, it is important for doctors in clinical departments other than internal medicine and pediatrics to consider FXIII deficiency and perform FXIII inhibitor testing when examining patients with prolonged bleeding of unknown cause or persistent bleeding after trauma.

## 1. Introduction

### 1.1. General Description of Factor XIII

Factor XIII (FXIII) deficiency is a rare but critical coagulopathy that affects people across a range of ages. FXIII deficiencies can be congenital or acquired. FXIII is critical in blood coagulation and plays a role in stabilizing blood clots by cross-linking fibrin molecules. Plasma FXIII has a tetrameric structure composed of two zymogen A subunits and two carrier/protective B-subunits that are non-covalently linked, forming a heterologous tetramer (FXIII-A_2_B_2_). FXIII is cleaved by thrombin, forming FXIII-A_2_* (active A subunit dimer in FXIII activated by thrombin plus Ca^2+^) and FXIII-B [1,2]. Although cellular FXIII-A_2_ is immunochemically identical to plasma FXIII-A_2_, it has different functions than plasma FXIII-A_2_. Cellular FXIII-A_2_ is located in platelets, megakaryocytes, monocytes, tissue macrophages, and the placenta [3,4].

### 1.2. Congenital FXIII Deficiency and Acquired FXIII Deficiency

#### 1.2.1. Congenital FXIII Deficiency

Congenital FXIII deficiency is an autosomal recessive disorder. The incidence of congenital FXIII deficiency is estimated to be 1 in 3–5 million births [5]. Congenital FXIII deficiency can result from mutations in A or B subunit genes, but disorders of A subunits are more common and severe [4,6]. Umbilical bleeding within a few days after birth occurs in 80% of cases of congenital FXIII deficiency. Intracranial hemorrhage, ecchymosis, hematomas, impaired wound healing, and prolonged bleeding after trauma are also typical symptoms [7]. Intracranial hemorrhage is the main cause of death in patients with congenital FXIII deficiency [5]. Women with congenital FXIII deficiency are unable to carry pregnancies to term and are susceptible to recurrent miscarriage [5,7]. The bleeding tendencies of patients with heterozygous FXIII-A deficiency (congenital FXIII deficiency caused by a mutation on one allele of the FXIII A subunit gene) are not yet completely understood. Spontaneous bleeding is not likely to occur in these patients; however, in conditions of hemostatic stress, such as trauma, surgery, and delivery, severe blood loss has been reported in a few cases [1]. No clear diagnostic criteria for congenital FXIII deficiency in Japan were identified during this review; however, the International Society on Thrombosis and Haemostasis has published an algorithm for the diagnosis and classification of FXIII deficiency [8]. Following these guidelines, clinicians who suspect FXIII deficiency based on clinical symptoms and family history may make a diagnosis following laboratory examination [8]. Routine laboratory tests such as clotting time or first screening tests, such as bleeding time, prothrombin time (PT), activated partial thromboplastin time (APTT), and platelet count, are normal in patients with congenital FXIII deficiency. Therefore, the first-line test for the diagnosis of FXIII deficiency is a quantitative functional FXIII activity test. If FXIII activity declines, further investigations are recommended for the detection and classification of FXIII deficiency. To classify FXIII deficiency, immunoassays are used to determine the plasma concentrations of FXIII-A_2_B_2_ complex, A and B subunit antigens, and the concentration of FXIII-A_2_ antigen in platelet lysate. If necessary, additional tests, such as evaluation of fibrin cross-linking by sodium dodecyl sulfate–polyacrylamide gel electrophoresis can be performed. Finally, the detection of the molecular genetic defects underlying FXIII deficiency is recommended [4,8]. The basic treatment for congenital FXIII deficiency is FXIII replacement, and this treatment is important to prevent critical bleeding, such as intracranial hemorrhage.

#### 1.2.2. Acquired FXIII Deficiency

Acquired FXIII deficiency is divided into two types: autoimmune and non-autoimmune. Autoimmune FXIII deficiency results from the generation of anti-FXIII antibodies, either neutralizing or non-neutralizing antibodies, and occurs in the elderly [5]. These antibodies have been described in association with systemic lupus erythematosus, autoimmune diseases, malignant diseases, and medications [8]. Non-autoimmune FXIII deficiency is caused by decreased synthesis and increased consumption of FXIII [5]. It appears that hematopoietic cells are responsible for the production of the A subunit, which is synthesized by the liver [3]. One report suggests that acquired FXIII deficiency is a relatively common phenomenon in patients following surgery or hospitalization in the ICU [9]. Most cases of acquired FXIII deficiency are not immune-mediated and are typically asymptomatic [10]. In contrast, acquired FXIII deficiency caused by anti-FXIII antibodies can be associated with critical bleeding complications. Bleeding symptoms of non-autoimmune FXIII deficiency are milder than those of autoimmune FXIII deficiency [3,5]. Most patients with autoimmune FXIII deficiency experience bleeding in soft tissues, such as intramuscular and subcutaneous tissues [5]. Autoimmune FXIII deficiency frequently causes bleeding in critical sites, e.g., intracranial and retroperitoneal bleeding [5]. In these patients, clots may form normally but begin to break down 24–48 h later because of weak cross-linking of fibrin, leading to subsequent episodes of bleeding [3]. Delayed bleeding can occur following surgery or trauma in FXIII-deficient patients. The diagnostic criteria for autoimmune FXIII deficiency with bleeding symptoms in Japan are shown in Table 1, and a flowchart of the diagnostic process is shown in Figure 1. There is no difference between the diagnostic criteria of ISTH and the guidelines in Japan [11]. Thus, it is necessary to perform immunological tests to measure FXIII inhibitors to make a definitive diagnosis of autoimmune FXIII deficiency. Treatment for autoimmune FXIII deficiency involves hemostatic therapy and immunosuppressive agents, such as steroids [12] The level of minimum FXIII activity required to prevent bleeding is unknown. Individuals with heterozygous congenital FXIII deficiency have increased bleeding propensity, with FXIII levels around 50%. In surgical settings, an FXIII target of 50% has been reported in a case series involving perioperative patients with congenital or acquired FXIII deficiency [10]. It has also been reported that 31% of FXIII activity levels are necessary for patients to remain asymptomatic [13].

Regarding prognosis, it has been reported that 73% of patients with autoimmune acquired FXIII deficiency recovered from the acute phase of bleeding symptoms, and 18% of patients died within one year of hospitalization or diagnosis [5]. Several cases of autoimmune acquired FXIII deficiency have been reported to relapse after clinical or functional remission [12].

### 1.3. Significance

We encountered a case of acquired FXIII deficiency that we report below. As stated above, it is necessary to perform immunological tests to measure FXIII inhibitors in order to diagnose FXIII deficiency. However, Japan has a limited number of facilities for performing immunological tests, and approval from the Institutional Review Board is required to perform these tests; thus, extensive time is required to obtain relevant test results. In particular, it is difficult to perform immunological tests at local hospitals with limited resources and without full-time hematology departments. In cases of critical bleeding, such as in the presented case, obtaining immunological test results can require several days or weeks, which may delay treatment. Delayed treatment may worse patient prognosis; therefore, it is necessary to initiate clinical examination and treatment in a timely manner. We performed a systematic review to investigate current practices and conditions for the diagnosis and treatment of FXIII deficiency, including the measurement of FXIII inhibitors in Japan.

## 2. Materials and Methods

### 2.1. Case Report and Systematic Review

#### 2.1.1. Case Report

We diagnosed and treated a patient with acquired FXIII deficiency in our hospital, a rural community hospital. The patient’s diagnosis, treatment, and clinical course are described below.

#### 2.1.2. Research Methods

This systematic review was prepared according to the preferred reporting items for systematic reviews and meta-analysis (PRISMA) guidelines. The study was registered on the international prospective register of systematic reviews (PROSPERO) platform, with Registration Number 298127.

#### 2.1.3. Systematic Review Data Sources

We searched for Japanese cases of FXIII on Ichushi Web and Google Scholar from 8 May 2021 to 18 July 2021. The words used in the search were “13 inshi ketsubosho”, “13 inshi kessonsho”, “XIII inshi ketsubosho”, “XIII inshi ketsubosho”, and “kotense ketsuyubyo 13”; all translate to “factor XIII deficiency” in English.

#### 2.1.4. Study Selection

The inclusion and exclusion criteria are listed in Table 2. We included case reports or case series executed in Japan that describe FXIII cases from onset to diagnosis. We excluded asymptomatic cases with FXIII deficiency, conference presentations, reports of anesthesia or surgery, and previously diagnosed cases. Results obtained in the Google Scholar search that overlapped with cases identified in the Ichushi Web search were excluded to avoid duplication.

#### 2.1.5. Data Extraction

We collected and analyzed the publication year, type of FXIII deficiency, age, sex, past history of autoimmune disease, past history of coagulation disorder, past history of malignancy, familial history of bleeding, sites of bleeding, laboratory data of PT, APTT, and FXIII activity, measurement of inhibitors such as mixing test and immunologic binding assays, the antigen of each subunit, measurement of FXIII activity of the patient, department of the first author, whether transfusion (red blood cells, fresh frozen plasma, and platelets) was required, treatment, and outcome in adopted cases. The evaluation methods for these variables are described below.

We categorized the cases into congenital and acquired FXIII deficiencies based on the authors’ diagnoses. However, we could not categorize acquired FXIII deficiency cases into autoimmune and non-autoimmune cases because some of the included reports did not provide sufficient evidence to determine whether the cases were autoimmune or non-autoimmune in nature.

We classified the sites of bleeding as intracranial, extremity, facial and truncal, thoracic cavity and mediastinum, abdominal cavity and retroperitoneum, surgical wound, traumatic wound, and others. We subdivided the extremities into subcutaneous bleeding, intramuscular bleeding, and intra-articular bleeding, and subdivided facial and truncal bleeding into subcutaneous bleeding and intramuscular bleeding. If there were multiple sites of bleeding, all were included as bleeding sites in the present analysis.

We categorized the treatment into treatment for primary disease, immunosuppression, and plasma-derived FXIII concentrate administration. Replacement therapy of FXIII included both temporary and continuous therapies because some case reports did not state when FXIII replacement was discontinued. We categorized the outcome as deterioration, including death, based on the report. Therefore, the period from diagnosis to the determined outcome was different in each case.

#### 2.1.6. Statistical Analysis

For continuous variables, the normality of the data was tested before applying the statistical tests. There was only nonparametric data, age; thus, we analyzed the data using the Mann–Whitney *U* test. Nominal variables were analyzed using Fisher’s exact test. All data analyses were performed using Easy R (version 1.54; R Foundation for Statistical Computing, Vienna, Austria). Statistical significance was defined as *p* < 0.05 [14].

## 3. Results

### 3.1. Case Report

A 74-year-old man presented with a one-month history of the right thigh and bilateral lower leg pain. He was in good health until two months prior to admission, when he developed pain in both legs after falling backward out of a chair. One month prior to admission, he reported a fever and difficulty in moving because of pain in his lower legs. He visited the emergency department of our hospital, and his fever resolved within a day, but the pain in both lower legs persisted. The pain gradually worsened until he was unable to traverse stairs. Three days before admission, he visited his family doctor with a chief complaint of pain and purpura of the lower legs and loss of appetite. He was prescribed painkillers, but his symptoms did not improve. Therefore, his family doctor referred him to the orthopedic department of our hospital, where the orthopedic doctor referred the patient to the internal medicine department because of anemia. His medical history included depression and insomnia, and he had no history of surgery. His current medications included ursodeoxycholic acid, biofermin, domperidone, and thiamine. At the time of admission, his vital signs were stable. He had edema, tenderness, rash, and purpura of the right thigh and bilateral lower legs. Computed tomography (CT) of the legs showed a hematoma in the right thigh and bilateral lower legs, but contrast CT did not show any extravasation (Figure 2). Magnetic resonance imaging of the lower legs revealed muscle inflammation in short T1 inversion recovery (Figure 3). Ultrasound of the bilateral lower leg showed a hematoma and inflammation in the muscles. Initial laboratory tests revealed normocytic anemia (hemoglobin, 6.8 g/dL; mean corpuscular volume, 102.6 fL) and acute kidney injury (creatinine, 1.28 mg/dL). Platelet count (130,000/μL), PT international normalized ratio (INR) (1.07), and APTT (26.2 s) were also normal. The results of the other laboratory tests are shown in Table 3. The orthopedic physician referred the patient to a general practitioner.

After admission, the patient required a transfusion of six units of red blood cells because of continuous anemia. We suspected coagulation abnormality, but PT-INR, APTT, and bleeding time were normal (1.0 s); thus, we suspected FXIII deficiency and examined FXIII activity. We diagnosed FXIII deficiency because FXIII activity was 47% as measured by the ammonia release assay. We performed a 1:1 close mixing test because we suspected acquired FXIII deficiency based on the patient’s age, no past history of bleeding, and no family history of bleeding. The patient was treated with 1 mg/kg/day of prednisolone (PSL), and his symptoms and anemia improved. After initiating treatment, we found that the close mixing test was negative. However, we continued treatment with PSL because of the late age of onset and efficacy of PSL treatment. We did not have a differential diagnosis that matched this clinical course, such as other autoimmune diseases. Considering that the close mixing test was negative, we suspected the presence of an antibody other than the neutralizing antibody. Following treatment, the intramuscular hematoma was improved, and the patient was able to walk as usual; thus, we decreased the PSL dose to 0.5 mg/kg/day. The patient’s symptoms improved and his FXIII activity was normal; thus, the amount of PSL was gradually reduced by 5 mg/month. Four months after initiating the PSL treatment, we reduced the PSL dose to 10 mg/day, and the patient’s condition did not worsen. 

### 3.2. Systematic Review

#### 3.2.1. Literature Selection

The study selection process is illustrated in Figure 4. The literature search identified 653 articles (507 articles from the Ichushi Web and 146 articles from Google Scholar). A total of 541 articles were excluded because they were not case reports or case series. A total of 39 articles were excluded based on the title and abstract, and 17 articles were excluded because they failed to meet the eligibility criteria. Finally, we identified 56 case reports and case series that included 64 patients from 1973 to 2021 (Table 4).

#### 3.2.2. Patients’ Background

There were 27 and 37 cases of congenital and acquired FXIII deficiency, respectively. Overall, 19 cases had secondary FXIII deficiency in the acquired group. The patients’ background is shown in Table 5.

The average age of these patients was 25.6 years (standard deviation (SD) = 26.0) and 54.8 years (SD = 26.4 years) in the congenital and acquired groups, respectively. Age was significantly different between the two groups. Sex, autoimmune disease, coagulation disorder, and malignancy were not significantly different between the two groups. Triggers of onset were divided into five groups: trauma, surgery and medical intervention, drug, other, and none. The most common trigger of onset was “none”, especially in the congenital group, and was significantly different between both groups (congenital, 77.8%; acquired, 32.4%). 

The number of patients in this review distributed across five-year periods is shown in Figure 5. The number of patients with congenital FXIII deficiency was higher than that of acquired FXIII deficiency before 2010, but this trend was reversed after 2010.

#### 3.2.3. The Frequency of FXIII Inhibitors, Subunits, and FXIII Activity

The frequency of FXIII inhibitors was 11.1% and 29.7% in the congenital and acquired groups, respectively, with no significant difference between the two groups. The frequency of the subunits was 37.0% and 13.5%, and the FXIII activity was 55.6% and 5.4% in the congenital and acquired groups, respectively. 

#### 3.2.4. The Categories of Physicians’ Specialties Detecting the Diseases

We defined the diagnosed department as the department of the first author. We divided the departments into internal medicine/pediatrics or others because we assumed that internal medicine and pediatrics are the primary departments that diagnose FXIII deficiency. Other major departments included surgery departments such as neurosurgery, orthopedics, obstetrics and gynecology, and plastic surgery.

Internal medicine and pediatrics were the diagnosing departments in 48.1% and 13.5% of congenital and acquired cases, respectively, with a significant difference between the groups (*p* < 0.01). The most common reporting department in the congenital group was pediatrics (33.3%), followed by neurosurgery (14.8%). Orthopedics (27.0%) was the most common reporting department in the acquired group, followed by obstetrics and gynecology (8.1%). Internal medicine, including hematology, was the reporting department for 8.1% of patients in the acquired group.

#### 3.2.5. The Patients’ Clinical Courses

##### Bleeding Sites

The patients’ bleeding sites are shown in Table 6. In patients with congenital FXIII deficiency, the most common bleeding sites were intracranial (22.2%), followed by facial and truncal (22.2%), and surgical and traumatic wounds (18.5%). In patients with acquired FXIII deficiency, the most common bleeding sites were surgical and traumatic wounds (54.1%), followed by extremities (21.6%), and abdominal cavity and retroperitoneum (10.8%). The rates of bleeding due to surgical and traumatic wounds were significantly different between the two groups (*p* < 0.01).

##### Treatment

We categorized the treatment into treatment for primary disease, immunosuppression, and plasma-derived FXIII concentrate administration. The frequency of plasma-derived FXIII concentrate administration was 70.4% and 81.1% in the congenital and acquired groups, respectively, with no significant difference between the groups (*p* = 0.38).

#### 3.2.6. Mortality

The outcome was defined as deterioration, including death. The frequency of deterioration or death was 7.4% and 8.1% in the congenital and acquired groups, respectively, with no significant difference between the groups (*p* = 1).

## 4. Discussion

This study revealed current practices of diagnosing and treating FXIII deficiency, including the measurement of FXIII inhibitors, used in Japan. According to the diagnostic criteria for diagnosis of autoimmune FXIII deficiency that are utilized in Japan (Table 1), measurement of FXIII inhibitors is necessary to diagnose acquired FXIII deficiency. However, this review revealed that only 29.7% of cases of acquired FXIII deficiency were diagnosed by the measurement of FXIII inhibitors. Considering the existence of secondary cases (19/37 cases (51.4%)), approximately 50% of acquired FXIII deficiency cases required the measurement of inhibitors in order to diagnose. Thus, measurements of FXIII inhibitors are not performed sufficiently under current practice conditions. The underlying reason for this insufficient testing may be that the criteria for diagnosing FXIII deficiency in Japan (Table 1) are not widely known. Since the diagnostic criteria are not well known, the necessity of FXIII inhibitor measurement itself may be poorly understood. Educating healthcare professionals on the importance of this diagnostic test could help improve testing rates. In addition, the procedure for inhibitor measurement is complicated, which may also contribute to the insufficient testing rate, especially in local hospitals without hematologists. Performing the necessary diagnostic tests prior to initiation of treatment is also time-intensive because it is necessary to first obtain the permission of the institutional ethics committee, request outsourced immunological testing, and submit samples by mail. In order for sufficient inhibitor measurement tests to be performed, it is necessary to simplify the testing method and clarify the diagnostic criteria for FXIII deficiency.

In this study, the mean age of FXIII deficiency onset was 25.6 years (median = 12 years) in congenital cases and 54.8 years in acquired cases. Cases of acquired FXIII deficiency identified in this literature review revealed several case reports that did not specify whether FXIII deficiency was autoimmune or non-autoimmune. In other studies, it has been reported that the median age of congenital FXIII deficiency diagnosis was 3.2 years [71], and the mean age of diagnosis reported for acquired FXIII deficiency cases, especially autoimmune cases, in a Japanese study was 70.1 years [5]. In this review, patients with congenital FXIII deficiency were, on average, older at the time of diagnosis than patients in previous reports. Among the cases analyzed, there were five cases of congenital FXIII deficiency diagnosed at 65 years or older. Three of these cases were diagnosed as congenital FXIII deficiency despite no previous history of bleeding tendency, and one was diagnosed as congenital based on the fact that the FXIII inhibitor tests were negative. This may have resulted in a higher median age of diagnosis. In such cases, it is necessary to confirm that the case is in fact congenital, not acquired. It should also be noted that there are forms of acquired FXIII deficiency that test negative for FXIII inhibitors. In this review, the mean age of acquired FXIII deficiency diagnosed was lower than that reported by a previous review of cases of autoimmune FXIII deficiency [5]. Since aging has been identified as a risk factor for autoimmune FXIII deficiency [5], the analysis of only autoimmune congenital FXIII deficiency cases may have resulted in a higher average age of diagnosis. On the other hand, this study also included acquired cases other than autoimmune cases, which may have resulted in a younger age of diagnosis compared with the previous analysis of only autoimmune cases.

Prior to 2010, the majority of case reports related to FXIII deficiency were congenital cases. The number of acquired cases increased after 2010, comprising the majority of FXIII deficiency diagnoses (Figure 5). This is most likely due to increased awareness of acquired FXIII deficiency due to a nationwide survey initiated by the Japanese Collaborative Research Group (JCRG) in Japan in April 2009. This group has distributed leaflets, questionnaires, and/or annual research report booklets on acquired FXIII deficiency to more than 2000 hospitals twice annually since 2009 [5]. Thus, the awareness of acquired FXIII deficiency has likely increased during this time, which may explain the increase in the number of diagnosed cases of acquired FXIII deficiency.

In congenital cases of FXIII deficiency, the most frequent bleeding sites identified were intracranial (22.2%), surgical wound and traumatic wound sites (18.5%), subcutaneous bleeds (18.5%), and intramuscular bleeds (14.8%). These bleeding sites were generally consistent with those noted in previous reports [1]. It has been reported that umbilical cord bleeding is seen in approximately 80% of congenital FXIII deficiency cases [1], but only 3.7% of the congenital FXIII deficiency cases identified in this review presented with umbilical cord bleeding. However, 48% of the congenital cases showed umbilical cord bleeding or bleeding from the umbilicus after umbilical cord shedding but prior to diagnosis; in most of the cases included in this review, FXIII deficiency diagnosis was not made at the time of umbilical cord bleeding. A history of persistent umbilical cord bleeding is frequent in patients with congenital FXIII deficiency and may be useful for screening by inquiry. In acquired cases, the frequent bleeding sites were surgical wounds and traumatic wounds (54.1%), subcutaneous (16.2%), and intramuscular (13.5%). Bleeding was often found in critical sites such as the intracranial region (8.1%) and abdominal cavity and retroperitoneum (10.8%).

There are no previous reports of common bleeding sites in patients with non-autoimmune acquired FXIII deficiency; however, it has been reported that the common sites of bleeding due to autoimmune FXIII deficiency are intramuscular (68%), subcutaneous (60%), intracranial (11%), abdominal and retroperitoneal (19%), and post-surgical sites (16%) [5]. Although the present review includes autoimmune and non-autoimmune acquired FXIII deficiency, the frequencies of intracranial hemorrhage, abdominal cavity bleeding, and retroperitoneal hemorrhage are not clearly different between the two. On the other hand, intramuscular and subcutaneous bleeding, which was considered to be frequent in the previous review [5], was less common according to the present review. In the present review, bleeding from surgical wounds and traumatic wounds was more frequent than in the previous review [5]. This may be because the present review included cases of non-autoimmune acquired FXIII deficiency, and non-autoimmune cases included cases in which FXIII levels decreased following surgery. In cases such as these, FXIII deficiency often presents with symptoms of unstoppable bleeding from a wound after trauma or surgery. Therefore, when PT and APTT are normal in a patient with such symptoms, it is important to consider FXIII deficiency as a differential diagnosis. In particular, acquired cases have significantly more bleeding from wound sites than congenital cases. Bleeding in cases of acquired FXIII deficiency is also common in the extremities. Since extremities are common sites of trauma, the frequency of bleeding from the extremities is high in acquired cases in which bleeding from wounds is likely to occur.

In congenital FXIII deficiency cases, the most common reporting department was pediatrics (33.3%), followed by neurosurgery (14.8%). Hematology reported only 3.7% of the cases included in this review. As FXIII deficiency symptoms appear in early childhood and intracranial hemorrhage often occurs in these patients, there are many reports from pediatrics and neurosurgery in congenital cases. In cases of acquired FXIII deficiency, hematology reported only 2.7% of cases, while the most common reporting departments were surgical departments, including orthopedics (27%), obstetrics and gynecology (8.1%), plastic surgery (5.4%), and neurosurgery (5.4%). This is likely due to frequent postoperative bleeding in cases of acquired FXIII deficiency. In addition, patients with acquired FXIII deficiency often bleed from the extremities (21.6%), explaining why the number of reports from orthopedics departments was the largest. Furthermore, it is possible that the reason why there are few reports by hematology departments is that FXIII deficiency is a well-known disease among hematologists; therefore, routine cases of FXIII deficiency presenting to hematology departments may not be published frequently. For this reason, although the main symptom of FXIII deficiency is bleeding, clinical departments other than internal medicine and pediatrics are often involved in medical treatment at the time of onset (congenital, 51.9%; acquired, 86.5%). Therefore, it is important for doctors in clinical departments other than internal medicine and pediatrics to consider FXIII deficiency when they examine patients with prolonged bleeding of unknown cause or persistent bleeding after trauma in order to make an earlier diagnosis. Thus, it is necessary to inform doctors other than internal medicine and pediatric practitioners about FXIII deficiency.

The outcomes of patients with congenital FXIII deficiency were improved or stable condition in 77.8% of cases and deterioration or death in 7.4%. The outcomes of acquired cases were improved or stable condition in 89.2% and deterioration or death in 8.1% of cases. The Japanese FXIII deficiency practice guideline states that 18% of patients with autoimmune acquired FXIII deficiency die within one year of hospitalization or diagnosis [4]. Other reports have suggested that the mortality rate of autoimmune acquired FXIII deficiency is 22% in Japanese people [1]. The patient outcomes identified in the present review are better than those reported by the Japanese FXIII deficiency practice guideline, although the present review includes non-autoimmune cases, and therefore, the previously reported outcomes are not directly comparable to those of this report. However, many of the case reports included in the present review only describe outcomes following the first treatment, and since patients were not followed after the first treatment, it is possible that death or relapses after treatment were not identified by this review.

Additionally, there are no large-scale studies that have examined the mortality rate of congenital FXIII deficiency. One study in Iran reported that the mortality rate was 15.4% [72]. It has been reported that the mortality rate of congenital FXIII deficiency is lower than autoimmune acquired FXIII deficiency. Once diagnosed, congenital FXIII deficiency is easy to treat by FXIII replacement. On the other hand, autoimmune acquired FXIII deficiency needs to be treated by immunosuppression [12]. This study showed no significant differences in outcomes between congenital and acquired FXIII deficiency. As mentioned above, many of the case reports in the present review only describe outcomes following the first treatment. Therefore, we believe that these outcomes may be due to the short follow-up periods and the inclusion of non-autoimmune acquired FXIII deficiency as acquired FXIII deficiency cases. Long-term follow-up studies are needed for further analysis.

Although the frequency of death was low in both congenital and acquired cases, FXIII deficiency causes life-threatening bleeding at critical sites such as the intracranial, thoracic, and abdominal cavities, and the retroperitoneum. Patients often experience repeated bleeding symptoms from birth in congenital cases, such as umbilical cord bleeding and uncontrollable bleeding in early childhood. In addition, compared with other congenital hemorrhagic diseases, FXIII deficiency causes more frequent intracranial bleeding [73] and umbilical cord bleeding [72]; therefore, early diagnosis is important for improving prognosis.

### Limitations

This systematic review had some limitations. Only cases reported in Japan were included in this review; therefore, ethnic or demographic differences may not be reflected. Some Japanese case reports may not have been included because they were published in English. Furthermore, differing diagnostic criteria may be set by different countries. Another limitation was that the diagnosis of some of the included cases may have been ambiguous concerning the nature of the FXIII deficiency (e.g., congenital or acquired, autoimmune or non-autoimmune). Finally, we did not follow up with cases after the drafting of this manuscript. Further studies are necessary to clarify the current state of FXIII deficiency diagnosis and treatment in Japan.

## 5. Conclusions

FXIII deficiency is a rare but serious condition. To diagnose acquired FXIII deficiency, it is necessary to perform immunological tests to measure levels of FXIII inhibitors. Timely examination and initiation of treatment are critical for improving the outcome of patients with FXIII deficiency. The present systematic review revealed that FXIII inhibitor measurement was only performed in 29.7% of acquired FXIII deficiency cases. Clinical departments other than internal medicine and pediatrics were often involved in the medical treatment of patients with FXIII deficiency at the time of onset (congenital, 51.9%; acquired, 86.5%). Acquired FXIII deficiency has become more widely known because of educational campaigns and surveys conducted by JCRG in Japan. It is important for doctors in clinical departments other than internal medicine and pediatrics to consider FXIII deficiency when examining patients with prolonged bleeding of unknown cause or persistent bleeding after trauma in order to diagnose this disease earlier. Thus, it is necessary to inform doctors other than internal medicine and pediatrics about this disease, especially regarding FXIII inhibitors in acquired cases.

## Figures and Tables

**Figure 1 jcm-11-01699-f001:**
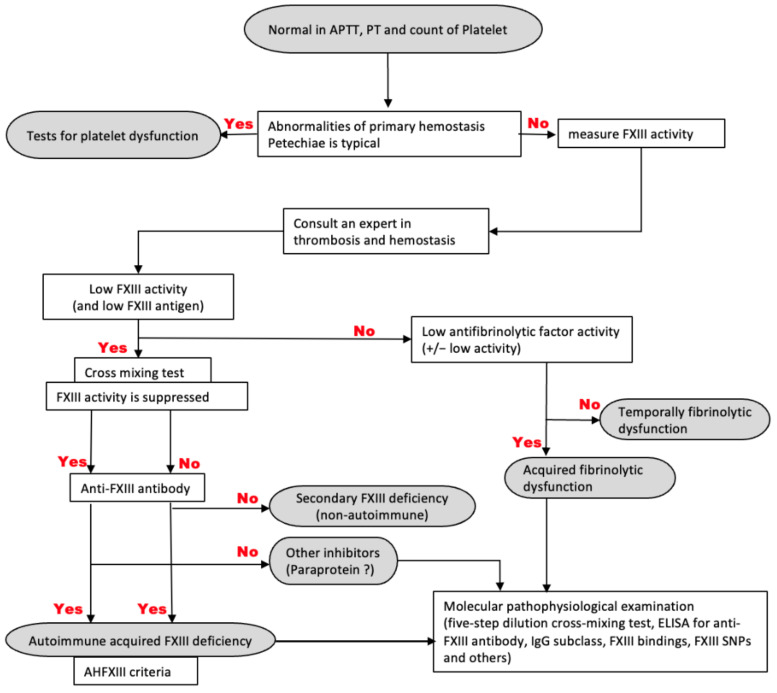
Flowchart for diagnosing autoimmune factor XIII deficiency with bleeding symptoms used in Japan. FXIII: factor XIII; ELISA: enzyme-linked immunosorbent assay; IgG: immunoglobulin G; SNPs: single nucleotide polymorphisms.

**Figure 2 jcm-11-01699-f002:**
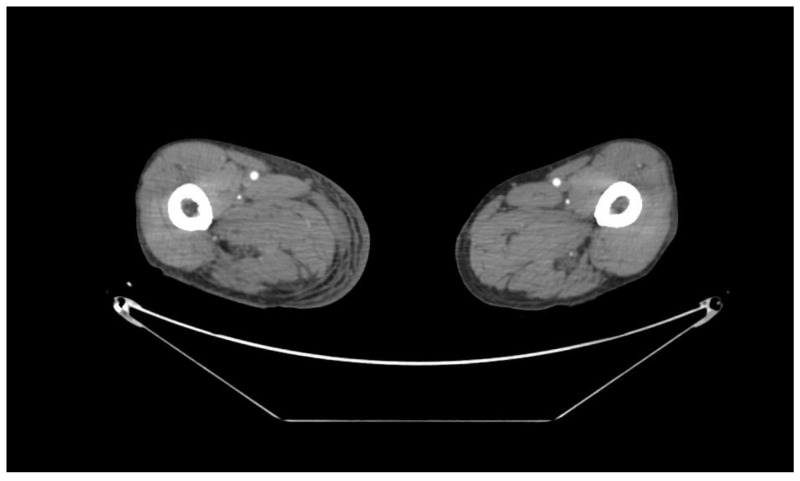
Contrast computed tomography of the bilateral thigh revealed hematoma of the right thigh and did not show any extravasation.

**Figure 3 jcm-11-01699-f003:**
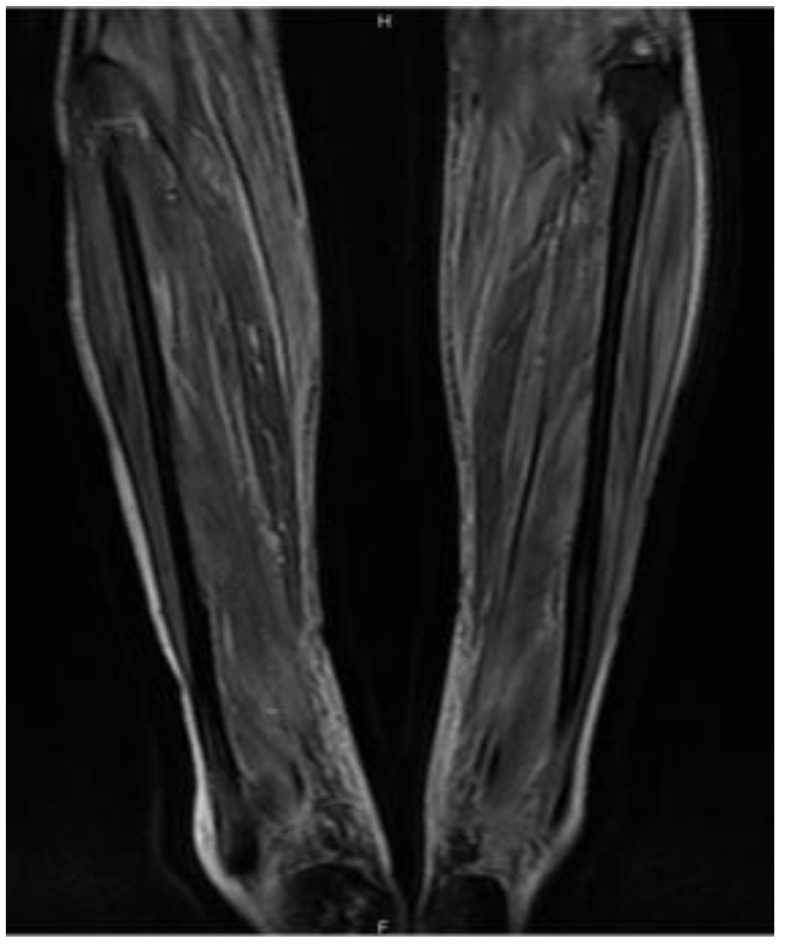
Magnetic resonance imaging of the lower leg revealed inflammation of the muscle in short T1 inversion recovery.

**Figure 4 jcm-11-01699-f004:**
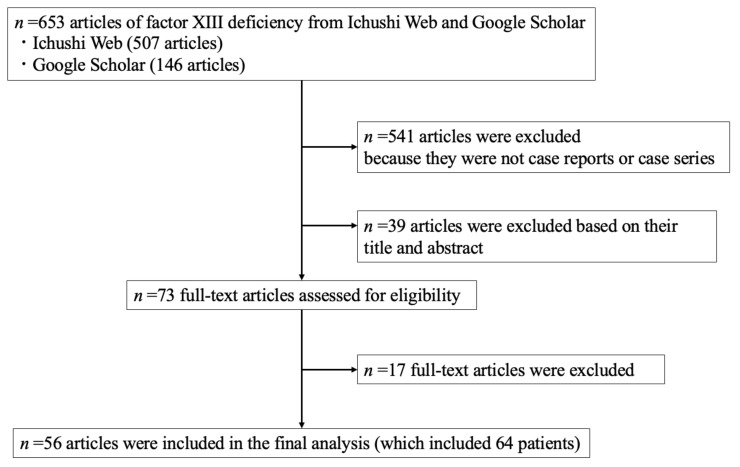
Study selection process.

**Figure 5 jcm-11-01699-f005:**
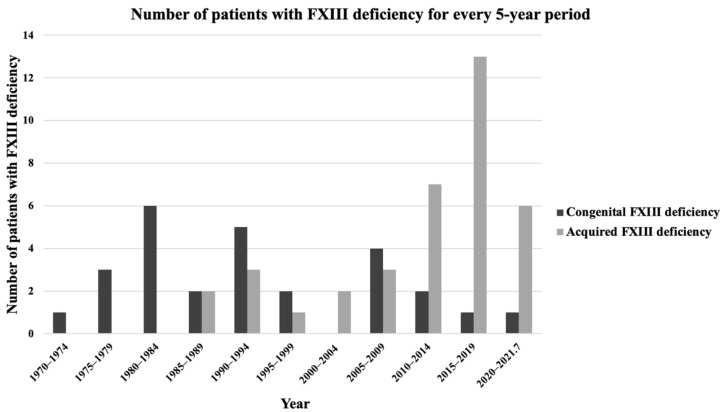
Distribution of factor XIII deficiency patients over time (5-year binned data). FXIII: factor XIII.

**Table 1 jcm-11-01699-t001:** The diagnostic criteria for autoimmune factor XIII deficiency with bleeding symptoms used in Japan [12].

Diagnostic Criteria for Autoimmune FXIII Deficiency Used in Japan
Possible criteria:
1. Recent onset of bleeding symptoms mainly in older adults;
2. No family history of congenital/inherited deficiency of FXIII or other coagulation factors;
3. Lack of previous bleeding symptoms especially in association with previous hemostatic challenges (e.g., surgery, invasive tests, trauma, etc.);
4. Not explained by excessive mediation such as anticoagulants and antiplatelet drugs;
5. Abnormality of FXIII parameter(s) on laboratory testing (FXIII activity and/or antigen <50%).
Probable criteria:
Criteria 1–5 plus the presence of FXIII inhibitors (positive by cross-mixing tests between the patient’s and healthy control’s plasma using standard function test after 2 h incubation at 37 °C).
Definitive criteria:
Criteria 1–5 plus the presence of anti-FXIII autoantibodies (positive by immunological methods).
FXIII: factor XIII

**Table 2 jcm-11-01699-t002:** Study inclusion and exclusion criteria.

Inclusion criteria	Case reports or case series Japanese cases Low FXIII activity observed on laboratory examination Reports describing an episode from onset to diagnosis
Exclusion criteria	Asymptomatic cases with FXIII deficiency Conference presentations Reports of anesthesia, surgery, and previously diagnosed FXIII deficiency cases

**Table 3 jcm-11-01699-t003:** Laboratory test results.

Complete Blood Count (CBC)					
White blood cell	4100	/μL	Red blood cell	1,960,000	/μL
Neutrophil	85.9	%	Reticulocyte	111,720	/μL
Lymphocyte	9.2	%	Hemoglobin	6.8	g/dL
Monocyte	4.0	%	Hematocrit	20.1	%
Eosinophil	0.4	%	Mean corpuscular volume	102.6	fL
Basophil	0.5	%	Platelet	130,000	/μL
**Coagulation**					
Prothrombin time-international normalized ratio		1.07			
Activated partial thromboplastin time		26.2	second		
D-dimer		6.50	μg/mL		
**Serum chemistries**					
Total protein	6.2	g/dL	Fe	25	μg/dL
Albumin	3.3	g/dL	Unsaturated iron-binding capacity	227	μg/dL
Total bilirubin	2.7	mg/dL	Ferritin	173.2	ng/mL
Direct bilirubin	0.9	mg/dL	Zn	54.7	μg/dL
Aspartate aminotransferase	31	IU/L	Cu	169	μg/dL
Alanine aminotransferase	12	IU/L	Vitamin B1	42	ng/mL
Alkaline phosphatase	33	U/L	Vitamin B12	≥1500	pg/mL
γ-glutamyl transpeptidase	17	IU/L	Folic acid	1.3	ng/mL
Blood urea nitrogen	27.6	mg/dL	Total homocysteine	29.0	nmol/mL
Creatinine	1.28	mg/dL	IgG	976	mg/dL
Na	138	mEq/L	IgM	80	mg/dL
K	3.1	mEq/L	IgA	159	mg/dL
Cl	101	mEq/L	Antinuclear antibodies	<40	times
Ca	8.2	mEq/dL	Proteinase-3-antineutrophil cytoplasmic antibodies	<1.0	U/mL
Creatine kinase	142	U/L	Myeloperoxidase-antineutrophil cytoplasmic antibodies	<1.0	U/mL
C-reactive protein	4.85	mEq/dL	Direct Coombs’ test	Negative	
Erythrocyte sedimentation rate	55	mm	Cold agglutinin	64	times
HBs antigen	0.01	IU/mL	Factor XIII activity	47	%
HBs antibody	<3.0	mIU/mL	Factor XIII inhibitor	Negative	
HBc antibody	5.55 (+)	S/CO			
Hepatitis B virus DNA	Negative	LogIU/mL			
Hepatitis C virus antibody	0.07	S/CO			
**Urine test**					
Leukocyte	(−)	Bilirubin		(−)	
Nitrite	(−)	Ketones		(−)	
Protein	(±)	Blood		(2+)	
Glucose	(−)	pH		5.5	
Urobilinogen	(1+)	Specific gravity		1.018	

**Table 4 jcm-11-01699-t004:** Included case reports in this review (56 case reports and case series which included 64 patients).

Case No.	Published Year	Reference	Age (years)	Sex	Type of FXIII Deficiency (C: Congenital, A: Acquired)	FXIII Inhibitor Measurement	Department of First Author
1	1973	[15]	12	M	C	−	Internal medicine
2	1976	[16]	18	F	C	−	Hematology
3	1977	[17]	6	F	C	−	Dental and oral surgery
4	1979	[18]	9	M	C	−	Pediatrics
5–1	1982	[19]	2	M	C	−	Pediatrics
5–2	1982	[19]	9	M	C	−	Pediatrics
6	1983	[20]	23	F	C	−	Internal medicine
7	1983	[21]	9	F	C	−	Neurosurgery
8–1	1984	[22]	1	F	C	−	Neurosurgery
8–2	1984	[22]	41	F	C	−	Neurosurgery
9	1985	[23]	0	M	C	−	Pediatrics
10	1988	[24]	19	F	A	−	Plastic surgery
11	1988	[25]	6	M	C	−	Pediatrics
12	1988	[26]	32	F	A	−	Obstetrics and gynecology
13	1991	[27]	58	M	A	+	Otorhinolaryngology
14	1991	[28]	62	M	A	+	Neurosurgery
15	1992	[29]	0	M	C	−	Pediatrics
16	1992	[30]	49	F	C	−	Anesthesiology
17	1993	[31]	4	F	C	−	Pediatrics
18	1993	[32]	66	F	A	−	Anesthesiology
19	1994	[33]	43	F	C	−	Obstetrics and gynecology
20	1994	[34]	22	M	C	−	Neurosurgery
21–1	1997	[35]	10	M	C	−	Pediatrics
21–2	1997	[35]	11	F	C	−	Pediatrics
22	1998	[36]	64	M	A	+	Dermatology
23	2002	[37]	7	F	A	−	Pediatrics
24	2002	[38]	8	M	A	−	Pediatrics
25	2005	[39]	0	F	C	−	Pediatric surgery
26–1	2005	[40]	9	M	A	−	Pharmacy
26–2	2005	[40]	10	F	A	−	Pharmacy
27	2007	[41]	69	M	C	−	Orthopedics
28	2008	[42]	69	M	C	−	Orthopedics
29	2009	[43]	67	F	A	−	Dermatology
30	2009	[44]	77	M	C	−	Clinical engineering
31	2010	[45]	51	F	C	+	Obstetrics and gynecology
32	2011	[46]	83	M	A	−	Orthopedics
33	2011	[47]	84	F	A	+	Orthopedics
34	2012	[48]	48	M	A	+	Orthopedics
35	2012	[49]	62	M	A	−	Orthopedics
36	2012	[50]	29	M	A	−	Orthopedics
37	2012	[51]	66	F	C	−	Anesthesiology
38	2013	[52]	62	F	A	−	Neurosurgery
39	2014	[53]	62	M	A	+	Internal medicine
40	2015	[54]	84	F	A	+	Urology
41	2015	[55]	61	F	A	−	Anesthesiology
42	2015	[56]	83	F	A	+	Pharmacy
43	2016	[57]	35	F	A	−	Obstetrics and gynecology
44	2016	[58]	87	M	A	−	Orthopedics
45	2017	[59]	75	F	A	−	Emergency
46–1	2017	[60]	29	F	A	−	Plastic surgery
46–2	2017	[60]	75	M	A	−	Hematology
47	2018	[61]	68	F	C	+	Emergency
48	2018	[62]	43	F	A	+	Obstetrics and gynecology
49	2018	[63]	16	M	A	+	Dental and oral surgery
50	2018	[64]	79	F	A	−	Cardiac surgery
51	2019	[65]	84	M	A	+	Hematology
52	2019	[66]	76	M	A	−	Urology
53–1	2020	[67]	76	M	A	−	Emergency
53–2	2020	[67]	48	F	A	−	Emergency
54	2020	[68]	19	M	A	−	Orthopedics
55	2020	[69]	15	F	C	+	Internal medicine
56–1	2021	[70]	58	F	A	−	Orthopedics
56–2	2021	[70]	88	F	A	−	Orthopedics
56–3	2021	[67,70]	81	F	A	−	Orthopedics

**Table 5 jcm-11-01699-t005:** Patients’ background characteristics.

		Congenital Factor XIII Deficiency (*n* = 27)	Acquired Factor XIII Deficiency (*n* = 37)	*p-*Value
Age (years), mean (SD)		25.6 (26.0)	54.8 (26.4)	<0.01
Sex, male, *n* (%)		12 (44.4)	17 (45.9)	1
Past history of autoimmune disease, *n* (%)		1 (3.7)	3 (8.1)	0.63
Past history of coagulation disorder, *n* (%)		5 (18.5)	1 (2.7)	0.07
Past history of malignancy, *n* (%)		0 (0.0)	4 (10.8)	0.13
Trigger of onset	Trauma, *n* (%)	1 (3.7)	5 (13.5)	0.39
	surgery, medical intervention, *n* (%)	5 (18.5)	11 (29.7)	0.39
	Drug, *n* (%)	0 (0.0)	3 (8.1)	0.26
	Other, *n* (%)	0 (0.0)	6 (16.2)	0.04
	None, *n* (%)	21 (77.8)	12 (32.4)	<0.01
Coagulation disorder, initial laboratory tests (prothrombin time, activated partial thromboplastin time), *n* (%)		4 (14.8)	5 (13.5)	1
Factor XIII activity ≤50%, *n* (%)		22 (81.5)	29 (78.4)	0.50
Measurement of factor XIII inhibitor, *n* (%)		3 (11.1)	11 (29.7)	0.13
Measurement of subunit, *n* (%)		10 (37.0)	5 (13.5)	0.04
Factor XIII activity test of parents, *n* (%)		15 (55.6)	2 (5.4)	<0.01
Diagnosing department	Internal medicine and pediatrics	13 (48.1)	5 (13.5)	<0.01
	Hematology	1 (3.7)	2 (5.4)	
	Internal medicine other than hematology	3 (11.1)	1 (2.7)	
	Pediatrics	9 (33.3)	2 (5.4)	
	Other, *n* (%)	14 (51.9)	32 (86.5)	<0.01
	Neurosurgery	4 (14.8)	2 (5.4)	
	Orthopedics	2 (7.4)	10 (27.0)	
	Obstetrics and gynecology	2 (7.4)	3 (8.1)	
	Plastic surgery	0 (0.0)	2 (5.4)	
Transfusion, *n* (%)		10 (37.0)	18 (48.6)	0.45
Plasma-derived factor XIII concentrate administration, *n* (%)		19 (70.4)	30 (81.1)	0.38
Immunosuppression, *n* (%)		0 (0.0)	6 (16.2)	0.04
Treatment of primary disease, *n* (%)		0 (0.0)	4 (10.8)	0.13
Outcome	Deterioration or death, *n* (%)	2 (7.4)	3 (8.1)	1

**Table 6 jcm-11-01699-t006:** Bleeding sites.

Bleeding Site		Congenital Factor XIII Deficiency (*n* = 27)	Acquired Factor XIII Deficiency (*n* = 37)	*p-*Value
Intracranial, *n* (%)		6 (22.2)	3 (8.1)	0.15
Extremity, *n* (%)		3 (11.1)	8 (21.6)	0.33
	Subcutaneous bleeding, *n* (%)	2 (7.4)	5 (13.5)	0.689
	Intramuscular bleeding, *n* (%)	1 (3.7)	4 (10.8)	0.387
	Intra-articular bleeding, *n* (%)	0 (0.0)	1 (2.7)	1
Facial and truncal, *n* (%)		6 (22.2)	2 (5.4)	0.06
	Subcutaneous bleeding, *n* (%)	3 (11.1)	1 (2.7)	0.302
	Intramuscular bleeding, *n* (%)	3 (11.1)	1 (2.7)	0.302
Thoracic cavity and mediastinum, *n* (%)		0 (0.0)	3 (8.1)	0.26
Abdominal cavity and retroperitoneum, *n* (%)		0 (0.0)	4 (10.8)	0.13
Surgical wound and traumatic wound, *n* (%)		5 (18.5)	20 (54.1)	<0.01
Other, *n* (%)		6 (22.2)	6 (16.2)	0.74

## Data Availability

The data presented in this study are available on request from the corresponding author.

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
