# Peer review of "Measuring Factor XIII Inhibitors in Patients with Factor XIII Deficiency: A Case Report and Systematic Review of Current Practices in Japan"

_jcm, 2022, doi:10.3390/jcm11061699_

Round 1

Reviewer 1 Report

Overall comment

Amano et al. present a comprehensive review over FXIII deficiency in Japan, with a focus on the diagnostic process of acquired FXIII deficiency, and a case report of acquired autoimmune FXIII deficiency. Diagnosis of FXIII deficiency remains a challenge and it is important to raise awareness among medical professionals. The article is well researched and well written and the topic is important. Some minor changes, as detailed below, are required.

Specific comments

  1. Page 1: The subtitle 1.1. seems not well suited to the content of the paragraph, which is rather a general description of FXIII than a summary on FXIII deficiency.
  2. Page1, line 39: „thrombin-truncated FXIII-A2“: It is recommended to use the nomenclature recommended by the FXIII ISTH SSC subcommittee (Muszbek et al. Factor XIII: recommended terms and abbreviations. J Thromb Haemost 2007;5:181-183).
  3. Page 2, lines 54-56: The definition of heterozygous FXIII deficiency written in bracket is not correct. These patients have a mutation on one allele only, and they usually have around 50% of normal FXIII activity and antigen levels.
  4. Page 3, lines 102-103: this very old notion that 5-10% of FXIII are sufficient to maintain haemostasis is obsolete, it should not be written any more.
  5. Page 3, line 105: as stated above in comment 3, heterozygous patients have typically FXIII levels around 50% (not greater).
  6. Page 8: For the patient described in the case report, FXIII activity was given in table 3 as 47%, but in the text (p. 8 line 226) it was stated as „lower than normal (40%)“. Which is the correct result? And how was FXIII activity measured (which assay)? Were FXIII A- and B-subunit antigen levels also measured?
  7. Page 11, table 5: What do the authors mean by „Autoimmune disease“ and „Coagulation disorder“ (parameters 3 and 4 in the table) – were these earlier diagnoses, i.e. a history of autoimmune disease and coagulation disorder?
  8. Page 13, paragraph on Mortality: it is surprising that there was no difference in mortality. Congenital FXIII deficiency is relatively easy to treat once diagnosed, due to the long half-life of FXIII, FXIII replacement therapy is only needed once every 4-6 weeks, it is efficient and well tolerated. Autoimmune FXIII deficiency on the other hand is very difficult to treat, as any administered FXIII is neutralised by the autoantibodies and immunesuppressive therapy is very aggressive and has a lot of complications. So the outcome and mortality should be much better in patients with congenital FXIII. Can the authors speculate on the possible reasons why they have seen no differences between the two groups?

Author Response

1. Page 1: The subtitle 1.1. seems not well suited to the content of the paragraph, which is rather a general description of FXIII than a summary on FXIII deficiency.

Response:We would like to thank the reviewer for this insightful comment. We agree with your excellent suggestion and have revised “1.1 Summary of factor XIII deficiency” to “1.1 General description of factor XIII” (line 34).

2. Page1, line 39: „thrombin-truncated FXIII-A2“: It is recommended to use the nomenclature recommended by the FXIII ISTH SSC subcommittee (Muszbek et al. Factor XIII: recommended terms and abbreviations. J Thromb Haemost 2007;5:181-183).

Response:We would like to thank the reviewer for this insightful comment. We agree with your excellent suggestion and have revised “thrombin-truncated FXIII-A2” to “FXIII-A2* (Active A subunit dimer in FXIII activated by thrombin plus Ca2+)” (lines 40-41).

3. Page 2, lines 54-56: The definition of heterozygous FXIII deficiency written in bracket is not correct. These patients have a mutation on one allele only, and they usually have around 50% of normal FXIII activity and antigen levels.

Response:We would like to thank the reviewer for this insightful comment. We agree with your excellent suggestion and have revised “Bleeding tendencies of patients with heterozygous FXIII-A deficiency (congenital FXIII deficiency caused by mutation of the FXIII A subunit gene) remain incompletely understood.” to “The bleeding tendencies of patients with heterozygous FXIII-A deficiency (congenital FXIII deficiency caused by a mutation on one allele of the FXIII A subunit gene) are not yet completely understood.” (lines 57-60).

4. Page 3, lines 102-103: this very old notion that 5-10% of FXIII are sufficient to maintain haemostasis is obsolete, it should not be written any more.

Response:We would like to thank the reviewer for this insightful comment. We agree with your excellent suggestion and have eliminated the sentence “an activity level above 5–10 % has been recommended to maintain hemostasis in patients with congenital FXIII deficiency”

5. Page 3, line 105: as stated above in comment 3, heterozygous patients have typically FXIII levels around 50% (not greater).

Response:We would like to thank the reviewer for this insightful comment. We agree with your excellent suggestion and have revised the statement “individuals with heterozygous congenital FXIII deficiency have increased bleeding propensity with FXIII levels greater than 50 %.” to “Individuals with heterozygous congenital FXIII deficiency have increased bleeding propensity with FXIII levels around 50 %.” (lines 107-109).

6. Page 8: For the patient described in the case report, FXIII activity was given in table 3 as 47%, but in the text (p. 8 line 226) it was stated as „lower than normal (40%)“. Which is the correct result? And how was FXIII activity measured (which assay)? Were FXIII A- and B-subunit antigen levels also measured?

Response:We would like to thank the reviewer for this insightful comment. We agree with your excellent suggestion and have revised “We diagnosed FXIII deficiency because FXIII activity was lower than normal (40 %).” to “We diagnosed FXIII deficiency because FXIII activity was 47% as measured by ammonia release assay.” (lines 243-244). Unfortunately, we did not measure FXIII A- and B- subunit antigen levels.

7. Page 11, table 5: What do the authors mean by „Autoimmune disease“ and „Coagulation disorder“ (parameters 3 and 4 in the table) – were these earlier diagnoses, i.e. a history of autoimmune disease and coagulation disorder?

Response:We would like to thank the reviewer for this insightful comment. We agree with your excellent suggestion. We have revised “We collected and analyzed the publication year, type of FXIII deficiency, age, sex, familial history of bleeding, sites of bleeding, laboratory data of PT, APTT, and FXIII activity, measurement of inhibitors such as mixing test and immunologic binding assays, antigen of each subunit, measurement of FXIII activity of the patient, department of first author, whether transfusion (red blood cells, fresh frozen plasma, and platelets) was required, treatment, and outcome in adopted cases.” to “We collected and analyzed the publication year, type of FXIII deficiency, age, sex, past histories of autoimmune diseases, coagulation disorders and malignancies, familial history of bleeding, sites of bleeding, laboratory data of PT, APTT, and FXIII activity, measurement of inhibitors such as mixing test and immunologic binding assays, antigen of each subunit, measurement of FXIII activity of the patient, department of first author, whether transfusion (red blood cells, fresh frozen plasma, and platelets) was required, treatment, and outcome in adopted cases.” (lines 170-176).

8. Page 13, paragraph on Mortality: it is surprising that there was no difference in mortality. Congenital FXIII deficiency is relatively easy to treat once diagnosed, due to the long half-life of FXIII, FXIII replacement therapy is only needed once every 4-6 weeks, it is efficient and well tolerated. Autoimmune FXIII deficiency on the other hand is very difficult to treat, as any administered FXIII is neutralised by the autoantibodies and immunesuppressive therapy is very aggressive and has a lot of complications. So the outcome and mortality should be much better in patients with congenital FXIII. Can the authors speculate on the possible reasons why they have seen no differences between the two groups?

Response:We would like to thank the reviewer for this insightful comment. We agree with your excellent suggestion and have added “Additionally, there are no large-scale studies which have examined the mortality rate of congenital FXIII deficiency. One study in Iran reported that the mortality rate was 15.4% [74]. It has been reported that the mortality rate of congenital FXIII deficiency is lower than autoimmune acquired FXIII deficiency. Once diagnosed, congenital FXIII deficiency is easy to treat by FXIII replacement. On the other hand, autoimmune acquired FXIII deficiency needs to be treated by immunosuppression [11]. This study showed no significant differences in outcomes between congenital and acquired FXIII deficiency. As mentioned above, many of the case reports in the present review only describe outcomes following the first treatment. Therefore, we believe that these outcomes may be due to the short follow-up periods and the inclusion of non-autoimmune acquired FXIII deficiency as acquired FXIII deficiency cases. Long term follow-up studies are needed for further analysis.” (lines 462-473).

Reviewer 2 Report

I highly appreciate the work undertaken by the authors to add to the rare disorder. The manuscript is interesting, however would like to have the following clarifications:

1: The table 1 explains the criteria for diagnosis of autoimmune Factor XIII deficiency with bleeding symptoms used in Japan. The authors should provide the reference to the body that provides these guidelines are taken. Also if they are different from the guidelines from ISTH, mention the main differences between the guidelines provided.

  1. Figure-1 Flowchart for diagnosing autoimmune factor XIII deficiency should be provided with the reference to the body providing the protocol.

The  flow chart lacks clarity, the authors may make it more presentable if feasible.

  1. In discussion para3 line 351, add the reference of figure 5. As per the figure 5 reporting about acquired cases has increased after 2010 as awareness program knocked the clinicians about it, but what would be the possible cause of such high acquired cases in Japan . 
  2. kindly provide an explanation about binned data from 2020-2024?

Author Response

1. The table 1 explains the criteria for diagnosis of autoimmune Factor XIII deficiency with bleeding symptoms used in Japan. The authors should provide the reference to the body that provides these guidelines are taken. Also if they are different from the guidelines from ISTH, mention the main differences between the guidelines provided.

Response:We would like to thank the reviewer for this insightful comment. We agree with your excellent suggestion and have revised “Table 1. The diagnostic criteria for autoimmune factor XIII deficiency with bleeding symptoms used in Japan” to “Table 1. The diagnostic criteria for autoimmune factor XIII deficiency with bleeding symptoms used in Japan [11]”. There is no difference between the diagnostic criteria of ISTH and the guidelines in Japan. (lines 103-104).

2. Figure-1 Flowchart for diagnosing autoimmune factor XIII deficiency should be provided with the reference to the body providing the protocol.

The  flow chart lacks clarity, the authors may make it more presentable if feasible.

Response:We would like to thank the reviewer for this insightful comment. We agree with your excellent suggestion and have revised Figure 1.

3. In discussion para3 line 351, add the reference of figure 5. As per the figure 5 reporting about acquired cases has increased after 2010 as awareness program knocked the clinicians about it, but what would be the possible cause of such high acquired cases in Japan.

Response:We would like to thank the reviewer for this insightful comment. We agree with your excellent suggestion and have revised “Prior to 2010, the majority of case reports of FXIII deficiency were congenital cases, while after 2010, the number of acquired cases increased, comprising the majority of FXIII deficiency diagnoses” to “Prior to 2010, the majority of case reports related to FXIII deficiency were congenital cases. The number of acquired cases increased after 2010, comprising the majority of FXIII deficiency diagnoses (Figure 5).” (lines 382-384).

This result could be due to increased awareness of acquired FXIII deficiency among Japanese physicians due to a nationwide survey and information provision regarding the diseases initiated by the Japanese Collaborative Research Group (JCRG) in Japan in April 2009.

4. kindly provide an explanation about binned data from 2020-2024?

Response:We would like to thank the reviewer for this insightful comment. We agree with your excellent suggestion and have revised Figure 5. Our search was completed in July 2021, so we revised “2020-2024” to ”2020-2021.7”.